# First Insights in the Relationship between Lower Limb Anatomy and Back Squat Performance in Resistance-Trained Males and Females

**DOI:** 10.3390/bioengineering10070865

**Published:** 2023-07-21

**Authors:** Céline Knopfli, Basil Achermann, Katja Oberhofer, Silvio R. Lorenzetti

**Affiliations:** 1Section Performance Sport, Swiss Federal Institute of Sport Magglingen (SFISM), 2532 Magglingen, Switzerland; celine.knopfli@hest.ethz.ch (C.K.); basil.achermann@baspo.admin.ch (B.A.); katja.oberhofer@kws.ch (K.O.); 2Department of Health Sciences and Technology (HEST), ETH Zurich, 8092 Zurich, Switzerland

**Keywords:** resistance training, velocity-based training, biomechanics, musculoskeletal anatomy

## Abstract

Identifying key criteria of squat performance is essential to avoiding injuries and optimizing strength training outcomes. To work towards this goal, this study aimed to assess the correlation between lower limb anatomy and back squat performance during a set-to-exhaustion in resistance-trained males and females. Optical motion captures of squat performance and data from magnetic resonance imaging (MRI) of the lower limbs were acquired in eight healthy participants (average: 28.4 years, four men, four women). It was hypothesized that there is a correlation between subject-specific musculoskeletal and squat-specific parameters. The results of our study indicate a high correlation between the summed volume of the hamstrings and quadriceps and squat depth normalized to thigh length (r = −0.86), and a high correlation between leg size and one-repetition maximum load (r = 0.81), respectively. Thereby, a marked difference was found in muscle volume and one-repetition maximum load between males and females, with a trend of females squatting deeper. The present study offers new insights for trainers and athletes for targeted musculoskeletal conditioning using the squat exercise. It can be inferred that greater muscle volume is essential to achieving enhanced power potential, and, consequently, a higher 1RM value, especially for female athletes that tend to squat deeper than their male counterparts.

## 1. Introduction

It is known that resistance training improves athlete performance by increasing muscle mass, power output, and maximum strength [1]. Thereby, the back squat is one of the most effective and powerful weight-bearing exercises, with wide-spread use across athlete conditioning for fitness and performance sports [2]. The intended use of the back squat is to exercise the musculature around the knee and hip joints and to build strength in the lower back. In particular, the knee extensors (e.g., vastus lateralis, vastus medialis, and rectus femoris) and hip extensors (e.g., biceps femoris, semitendinosus, and gluteus maximus) are regarded as the significant actors during the back squat, while other muscles, such as the soleus and erector spinae, additionally act to stabilize the joints [3].

Identifying key musculoskeletal and biomechanical criteria that impact squatting performance is essential to avoiding injuries while ensuring maximum training efficiency and has generated a great deal of interest in researchers in recent years [3,4,5,6]. Here, squat depth has played a crucial role in defining different technical recommendations for athlete performance and rehabilitation programs [2,3,7]. In particular, a significant correlation has been shown between squat depth, the lifted load, and the relative muscular effort in the ankle plantar flexors and hip and knee extensors [8]; similar average joint ranges of motion in the lower limbs and pelvis were found during the full back squat in healthy, recreationally active, college-aged volunteers (twenty female, four male) [3]. Notably, the so-called full or deep back squat has been defined so that the thighs drop below the horizontal line in the sagittal plane until the thigh touches the calf and the body cannot go lower; meanwhile, for the half or parallel back squat, the downward eccentric phase is expected to terminate when the thigh reaches the horizontal line [4].

It has been further suggested that peak lifting velocity could be a valuable performance criterion of an athlete’s relative power output ability and speed during the one-repetition maximum (1RM) back squat [9]. Lifting velocity is generally measured during so-called velocity-based training (VBT) as a promising objective approach to directly target and quantify strength training intensity. VBT is based on the finding that a linear relationship exists between lifting velocity and load intensity, which allows the prediction of the 1RM as valid indicator of maximal force capacity without the athlete taking the risk of an injury with high loading [6,10]. It was recently shown in 20 strength-trained males that the relative loss in lifting velocity during a set-to-exhaustion of full back squats is strongly related to the repetitions left in reserve, thus serving as a good estimate of muscular fatigue [11]. Furthermore, peak lifting velocity was found to correlate with relative peak power during the 1RM back squat in 21 male college athletes; yet, no correlations were found between peak lifting velocity and training age or femur length in the same study group [9].

Looking at the trajectory of the velocity during the concentric phase of the back squat, the so-called sticking region (Figure 1) has been identified as the weakest and most constraining element of squat performance, as it has a profound impact on the load that the athlete is able to lift [11]. The sticking region is the region during the concentric phase in which a loss in lifting velocity and a decrease in vertical acceleration occurs and has been shown to be associated with changes in the activation timing of the knee and hip extensors with increasing neuromuscular fatigue [12,13,14,15,16]. In particular, results from several studies in recreationally trained male lifters indicate that decreased muscle activity of the quadriceps occurs around the sticking region, with increased activation of the hamstrings and glutei muscles needed to get the athlete past this critical movement phase towards success versus failure [5,16,17].

Finally, muscle force potential, or muscle strength, was pointed out as a key criterion for accurately monitoring and prescribing squat technique [18,19]. Interestingly, it was recently shown by means of magnetic resonance imaging (MRI) and isometric strength testing in 52 healthy young men that quadriceps femoris muscle volume served as a better predictor of knee extensor force and muscular strength compared to muscle cross-sectional area (CSA) [20]. The statistically weaker relationship of CSA with muscle force compared to muscle volume contradicts the classical physiological theory that CSA best reflects the number of actin–myosin cross bridges that generate tensional force between muscle origin and insertion. Yet, earlier studies have already demonstrated that lateral force transmission should be considered [21], suggesting that muscle volume is a better determinant of muscle force potential compared to CSA. Muscle force potential is determined by muscle mass and, thus, muscle volume. Fukunaga et al. [22] suggested that lower limb muscle volume should be studied in more detail to research subject-specific differences in strength training performance outcomes.

In summary, the analysis of key musculoskeletal and biomechanical criteria to monitor squatting performance is an active research area across rehabilitation and recreational and performance sports. Here, only limited insights have so far been gained into the correlation between subject-specific musculoskeletal anatomy and squat performance, despite the correlation being considered critical for defining targeted exercise recommendations to ensure training safety and efficiency [2]. Most recently, it was found that the correlation between anthropometric variables and squat performance was gender-specific due to differences in fat percentage, lean body mass, and thigh length between resistance-trained males and females [4]. In particular, thigh length was found to be inversely correlated with squat performance in males (*n* = 19), while females (*n* = 17) displayed an inverse relationship between fat percentage and squat performance, respectively. Further evaluation of these anatomical parameters in relation to exercise performance in more participants and different study groups was considered beneficial when utilizing the back squat for specific strength training protocols.

Towards this goal, the main objective of the present pilot study was to find out whether there exists a correlation between subject-specific lower limb anatomy (i.e., hamstrings and quadriceps muscle volume, lower limb length, and lower limb size) and key squat-specific performance criteria (i.e., 1RM back squat load, squat depth, concentric lifting velocity, and existence of a sticking region) during a set-to-exhaustion in healthy, resistance-trained males and females. Such correlations were already found in VBT [6,10].

Therefore, it was hypothesized that thigh muscle volume as a measure of muscle force potential, assessed by means of MRI, correlates with squat lifting velocity and subject-specific 1RM. Furthermore, it was hypothesized that the thigh length is inversely correlated to squat lifting velocity, with differences in the correlation expected between males and females similar to the findings of Falch et al. [4]. Additionally, it was hypothesized that there is an association between lower limb anatomy and the presence or absence of a sticking region [22], whereby the ratio between the length and the circumference of thigh versus shank were considered to play a decisive role.

As a secondary objective, the differences in anatomical and squat-specific parameters were compared between female and male participants to provide preliminary insights into potential differences between genders. The novelty and main contribution of the present work was expected to result from the extension of analyses to consider MRI-based muscle volume as an indicator of squat performance, as well as from the analysis of the sticking region in relation to lower limb anatomy during VBT.

## 2. Materials and Methods

### 2.1. Data Collection

A total of 8 healthy, recreationally active participants were recruited for this study (*n* = 8, 4 males, 4 females). The inclusion criteria were to be healthy and have one year of strength training experience with at least one training for one hour per week. Exclusion criteria included musculoskeletal complaints, including pain or injury of the lower limbs. The participants had to fill out a questionnaire about the in/exclusion criteria and experience. The questionnaire was evaluated by the same trained examiner. All participants were familiar with the squat exercise and comfortable with the study protocol, including the use of MRI. Ethical approval for this study was given by the regional ethics committee (Kantonale Ethikkommission Bern, Switzerland, Projekt-ID: 2021-00403, approved 7 February 2022) and all participants gave their written informed consent prior to data collection. Data collection and analysis for this study took place between September 2022 and February 2023 and included one appointment for the assessment of anthropometric measures and squat performance measures and a second appointment for MRI of the lower limbs.

The data collection protocol is further outlined in the following. A standard rig with safety drop boxes, a barbell, and weight plates according to the International Weightlifting Federation (IWF) norms was used for squat performance testing (Figure 2). To assess squat depth and lifting velocity, a Vicon Motion System (Vantage 5; Vicon Motion Systems Ltd., Oxford, UK) was used with ten infrared cameras placed around the rig and two reflective markers placed on either end of the barbell to record the barbell displacement (Figure 2). The Vicon cameras were operated by an Antec WorkBoy desktop (Antec, Taipei, Taiwan) with Vicon Nexus software (Version 2.9; Vicon Motion Systems Ltd., Oxford, UK). The system captured the data with a sampling rate of 100 Hz.

Prior to data acquisition, the participants completed an individual warm-up session of 5–10 min and a set of squats without weights to familiarize themselves with the training equipment and test protocol. The squat performance testing protocol was based on the most current knowledge in VBT with clear instructions given by a trained coach on the order of tests and how to execute the free weight back squat with maximum voluntary lifting velocity [10,11]. VBT recommendations were adopted as an objective approach to assess squat performance by means of velocity measures during exercise execution.

The squat performance testing protocol comprised three parts: (1) incremental loading to establish each individual’s load–velocity profile (5 × 3 repetitions starting with 45% of the estimated 1RM with 10% load increments and a 3 min pause between sets), (2) a 1RM test with self-selected weight increments and an individual pause between attempts until failure, and (3) a set-to-exhaustion at 80% of the determined 1RM with maximum voluntary lifting velocity according to VBT and individual repetitions until failure. For all tests, participants were instructed to descend in a controlled downward movement at least until they passed the parallel squat position (knee flexion angle below parallel in the sagittal plane). Once they passed the parallel squat position, participants were free to descend further into their preferred squat position before executing the concentric movement phase of the squat with maximal voluntary lifting velocity. Participants were allowed to wear their standard training equipment (e.g., knee pads and belts) and chose their individual stance widths and hand positions on the barbell. During each set, the participants were verbally motivated to perform the lifts with maximum voluntary velocity and best performance level. A 10 min break was taken between the 1RM test and the set-to-exhaustion.

Anthropometric data, including size, weight, and leg length and circumferences, were measured by a trained practitioner according to the standards of the International Society for the Advancement of Kinanthropometry (ISAK). Measurements were taken following the squat performance testing and each measure was taken at least twice to be averaged. Furthermore, MRI data of the lower limbs of all participants were obtained using a 3 Tesla Siemens Magnetom Prisma scanner, with the participant in a supine position with neutrally extended legs. MRI data acquisition was conducted by a trained radiographer, with the images ranging from the upper iliac crest to the feet with the scan parameters being: 96 slices each, voxel size = 0.65 × 0.65 × 3.0 mm^3^, TR/TE1/TE2 = 3.9/1.23/2.46 ms, FOV = 445 × 418 mm^2^, fat- and water-separating reconstruction.

### 2.2. Data Analysis

Squat depth and lifting velocity were calculated for each repetition of the set-to-exhaustion based on the position data of the reflective markers on the barbell from the optical motion capture. Thereby, squat depth was determined from the vertical deflection during the eccentric phase of the movement, while lifting velocity was determined as the time derivative of the vertical deflection during the concentric movement phase. The presence or absence of a sticking region (Figure 1) was assessed based on the lifting velocity during the last repetition of the set-to-exhaustion according to the definition proposed by van den Tillaar, Andersen, and Saeterbakken [15] and recorded as a nominal variable (i.e., 0 or 1).

The open-source software SASHIMI (Version 1.2.0.0; MATLAB) was utilized to segment the hamstrings and quadriceps muscles from the MRI. A first contour of the target muscles (i.e., biceps femoris, semimembranosus, semitendinosus, rectus femoris, and the vasti group) was manually defined for a layer of images in the midportion of the long axis of the thigh, using previously segmented images as guidelines [23]. Then, an automatic tracking function in SASHIMI was applied to segment all other images. The resulting point clouds (i.e., muscle boundaries) were reviewed by a trained medical practitioner, and, if needed, adjusted manually to ensure correct segmentation of the target muscles. Muscle volume was calculated based on the segmented data using the so-called cylinder method, considered the gold standard [24]. Thereby, muscle volume (*V*) was determined as the summation of straight cylinders between sequential MRI slices with slice spacing (h) and the respective muscle cross-section areas (*CSA_i_*):$$V=\sum_{1}^{n}CSA_{i}*h$$

The programming language Python 3.9 was used for all data analyses, including the processing of the data from the optical motion capture, the derivation of muscle volume based on the segmented data from MRI, as well as the statistical analysis.

### 2.3. Statistical Analysis

Statistical analysis was conducted with Python 3.9 analysis scripting language. Squat depth was normalized to individual thigh length for statistical comparison between participants. Prior to statistical analysis, muscle volume, normalized squat depth, and concentric lifting velocity were checked for normal distribution with the Shapiro–Wilk test. Spearman correlation analysis and Pearson correlation analysis were used for correlation analyses of normally and non-normally distributed parameters, respectively.

Correlation analysis was conducted to investigate the relationship between the anatomical parameters of the lower limbs (i.e., summed hamstrings and quadriceps muscle volumes, hamstrings to quadriceps volume ratio, leg length, leg length ratio, leg size, and leg size ratio) and squat-specific parameters (i.e., 1RM back squat load, normalized squat depth, mean squat depth, average concentric velocity, peak concentric velocity, and existence of a sticking region). Leg length was calculated as the sum of the thigh and shank length, leg length ratio as the ratio between the thigh and shank length, leg size as the sum of the length times the circumference of the thigh and shank, and leg size ratio as the ratio between the length times the circumference of the thigh versus shank, respectively. Following Mukaka [25], correlation coefficients (r) were interpreted as low (r = ±0.3–0.5), moderate (r = ±0.5–0.7), or high (r = ±0.7–0.9), respectively.

Moreover, the percentage disparity in mean outcome parameters between males and females was derived for a preliminary assessment of the differences in anatomical and squat-specific outcome parameters between genders. Due to the low sample size (4 females, 4 males), gender-specific differences were not assessed for statistical significance but merely analyzed to identify trends that may help guide the direction of future research.

## 3. Results

A total of eight participants (four women and four men) signed up for this project. They met all inclusion criteria, signed informed consent, and enrolled in the study. Participants’ characteristics, including anthropometric measures and muscle volumes from MRI, are given in Table 1.

The determined average 1RM back squat load was 141.6 (36.8 kg). The participants completed on average 9.75 (4.4) repetitions during the set-to-exhaustion at 80% 1RM. The musculoskeletal and squat-specific outcome parameters are given in Table 1, including the percentage difference between male and female participants. Except for the average concentric velocity, the pooled data across participants were not normally distributed. Thus, Spearman correlation analyses were used.

The average 1RM back squat load was 141.6 kg (SD: 36.8 kg, Range: 95–190 kg). Thereby, the 1RM load was clearly different between male and female participants (Table 1). The 1RM back squat load across the pooled data was highly correlated to leg size (r = 0.81, *p* = 0.015), with a moderate correlation also found to the hamstrings and quadriceps muscle volume (Figure 3a and Figure 4). Leg size was calculated as the circumference times the length of the thigh and shank.

The mean absolute back squat depth during the set-to-exhaustion was 64.44 cm (SD: 6.54, Range: 56.00–71.63 mm), and the mean back squat depth normalized to thigh length was 1.436 (SD: 0.154, Range: 1.12–1.6), respectively. A moderate negative correlation was found between the back squat depth and the summed hamstrings and quadriceps muscle volumes across participants (r = −0.57, *p* = 0.14), with a higher correlation revealed between these two parameters by normalizing squat depth to thigh length (i.e., r = −0.86, *p* = 0.01). No further moderate or high correlation was observed between back squat depth, as well as normalized back squat depth, and any other musculoskeletal parameter (Figure 4). Yet, interestingly, a slight difference was found in the normalized back squat depth between male and female participants, with females squatting overall deeper (Table 1).

The average mean concentric lifting velocity during the set-to-exhaustion was 0.430 m/s (SD: 0.039, Range: 0.367–0.475 m/s), while the average peak lifting velocity was 1.366 m/s (SD: 0.365, Range: 0.846–1.77 m/s), respectively. Thereby, a moderate negative correlation was found between the average concentric velocity and the leg length ratio across participants (r = −0.56, *p* = 0.15), while a slightly higher moderate negative correlation was found between the peak concentric lifting velocity and the leg length ratio (r = −0.69, *p* = 0.06), respectively (Figure 3c,d and Figure 4).

A clear sticking region during the last repetition of the set-to-exhaustion was present in six out of eight participants. The lifting velocity prior to the sticking region reached an average peak of 0.385 +/− 0.116 m/s and then dropped to an average minimum of 0.188 +/− 0.067 m/s, followed by a fast increase to a maximum of 0.734 +/− 0.256 m/s. The two participants without clear sticking regions exhibited a gradual increase in their movement’s velocity during the concentric movement phase. Interestingly, a high correlation was found between the existence of a sticking region and the leg size ratio (r = 0.7, *p* = 0.1).

## 4. Discussion

The goal of the present study was to assess the correlation between lower limb anatomy and back squat performance during a set-to-exhaustion in resistance-trained males and females. The participants completed on average 9.75 (4.4) repetitions during a set-to-exhaustion at 80% 1RM, with the actual 1RM back squat load determined to be 141.6 kg (36.8 kg). The key findings from the present study indicate a high negative correlation between the summed hamstrings and quadriceps muscle volumes and the back squat depth normalized by thigh length (r = −0.86, *p* = 0.006), as well as a high positive correlation between leg size and the 1RM back squat load (r = 0.81, *p* = 0.015), respectively.

Despite the low sample size, the authors consider it noteworthy to look closer at the observed differences between males and females in comparison to existing knowledge in the literature. As expected, the 1RM back squat of the males was larger compared to the females, which is likely attributed to their larger muscle mass. Indeed, the 1RM back squat load across the pooled data was highly correlated to individual leg size (Figure 3a), with a moderate positive correlation also found between the 1RM and the summed quadriceps and hamstrings muscle volume (Figure 4). Based on this finding, it can be inferred that greater muscle volume is essential to achieve enhanced power potential and, consequently, a higher 1RM value. Interestingly, recent research reported a correlation between muscle mass and squat depth in female athletes (*n* = 7), whereby performance outcome measures for the parallel or half squat largely depended on the cross sectional area of the vastii muscles, but not the hamstrings [26]. Here, it would be interesting to further investigate whether similar correlations are found across female-only, male-only, and mixed-gender study groups, including participants with a large variation in muscle mass and 1RM performance.

Although all participants were asked to perform a full squat (knee flexion angle below parallel in the sagittal plane), minor variations in technique were observed due to variations in their primary sport and training experience. Here, a high negative correlation was found between normalized squat depth and summed muscle volume (r = −0.86, *p* = 0.006), with a trend of females squatting deeper than males. Gender-specific differences in squat depth were previously also reported in fifty-eight healthy volunteers during the single leg squat, suggesting different motor strategies at all levels of the kinematic chain between males and females [3]. Therefore, the hamstrings and quadriceps muscles contribute to knee joint stabilization during the single- and double-leg squat, and thus, play a critical role in executing different squat variations correctly. Nevertheless, subject-specific muscle recruitment patterns and vertical force production are known to be different between the double- and the single-leg squat [22], with gender-specific differences in lower limb anatomy likely contributing to performance outcomes. Additionally, squat depth in the present work was derived from barbell deflection in line with the state-of-the-art technology also used in VBT [6,10,11]. Individual differences in joint kinematics (i.e., knee, hip, and trunk position) that may have biased vertical squat depth measures were not taken into account. Further research into the relationship between subject-specific anatomy, musculoskeletal kinematics, and squat-specific outcome measures is advisable.

The present study protocol was defined in line with VBT recommendations as an objective method to monitor strength training performance. A moderate negative correlation was found between the average, as well as the peak, concentric lifting velocity and thigh-to-shank length ratio (Figure 4), while no correlation was found between average or peak lifting velocity and muscle volume across the pooled data. This finding is surprising and indicates that skeletal dimensions may be more indicative of an athlete’s ability to lift weight as fast as possible than muscle size. Particularly, a higher thigh versus shank length may indicate optimized lever arms with higher acceleration potential of the key muscle groups involved in the squat exercise. Previous research reported a correlation between peak velocity and maximum power output in a back squat study involving twenty-one college-aged male adults; yet, this research did not find a correlation between peak lifting velocity and femur length, which seems to contradict our findings [9]. However, it needs to be pointed out that the present study group was small with *n* = 8, and all results have to be discussed and interpreted carefully. Differences between study groups in training experience and movement execution technique may have influenced performance output parameters, particularly in the context of VBT. As a next step, it would be interesting to assess the correlation between muscle volume and lifting velocity at lower loads or during ballistic tasks to gain more decisive insights into skeletal dimensions versus muscle volume as determinants of speed. Hopefully, the present study can thereby encourage and point future research in this direction.

Previous research suggested that the sticking region (Figure 1) is the weakest and most constraining element of squat performance [11]. Here, we found that the presence of a sticking region moderately correlated to the thigh to shank size ratio (r = 0.63, *p* = 0.1). From this finding, it may be inferred that there is a possible link between lower limb anatomy and lifting technique, which has also been suggested in previous work [13,15]. Particularly, the sticking region was associated with low muscle force output and altered coordination between hip, knee, and ankle joint movements, with indications that the involved muscles may be at a disadvantageous position (i.e., length or moment arm) for force production during this critical movement phase. Importantly, the present study offers the first investigation into the potential correlation between lower limb anatomy and the presence of a sticking region during the back squat exercise, and further investigations in this direction are highly recommended.

Analyzing muscle volume instead of muscle mass was previously suggested to be advantageous in understanding the relationship between muscle volume, force, and acceleration [20,21], as it takes into account transmission forces between fibers. Nonetheless, it is noteworthy to point out that this study only considered hamstrings and quadriceps muscle volumes in a small study group (*n* = 8), without taking into account other muscles or fat mass, which may have influenced the study outcomes. Furthermore, it is noteworthy to point out that the cylinder method used to estimate muscle volume and leg size may have introduced a potential source of error due to incorrect identification or segmentation of the target muscles. Thus, it should be acknowledged that the chosen metric of outcome parameters is not conclusive. MRI is established as a method to assess soft tissues in a medical environment.

The main limitation of the present study was the low sample size of *n* = 8, which necessitates care in the interpretation of findings and drawing of preliminary conclusions, especially regarding the resulting differences between genders. Nevertheless, the present sample size is in line with similar studies in the literature that have analyzed muscle architecture in relation to squat performance in vivo [26]. In particular, the acquisition and analysis of MRI data is time-consuming, complicated, and costly, thus posing challenges for including more participants. Therefore, there are not yet that many studies published in the area of sport sciences using this technology. The quantification of muscle volume using MRI also has limitations and depends on the position of the human body [27]. Here, the primary goal of the present study was to provide insights into the relationship between lower limb anatomy and back squat performance across the entire study group, with the analysis of differences between genders being secondary. Further work is needed to confirm the present results and make statistically valid conclusions.

## 5. Conclusions

In summary, we have established preliminary associations between lower limb anatomy and various parameters specific to the back squat exercise. Our findings suggest a high negative correlation between muscle volume and normalized back squat depth, as well as a high positive correlation between leg size and 1RM back squat load, respectively. Based on the present findings, it can be inferred that greater muscle volume is essential to achieve enhanced power potential, and, consequently, a higher 1RM value, especially for female athletes that tend to squat deeper than male counterparts. The results from this study provide novel insights for coaches and athletes to facilitate targeted musculoskeletal conditioning using the squat exercise. To derive more comprehensive conclusions, it is advisable to conduct further research with more participants, particularly analyzing gender-specific differences in muscle activity patterns, movement kinematics, and exercise performance through biomechanical analysis of different squat techniques.

## Figures and Tables

**Figure 1 bioengineering-10-00865-f001:**
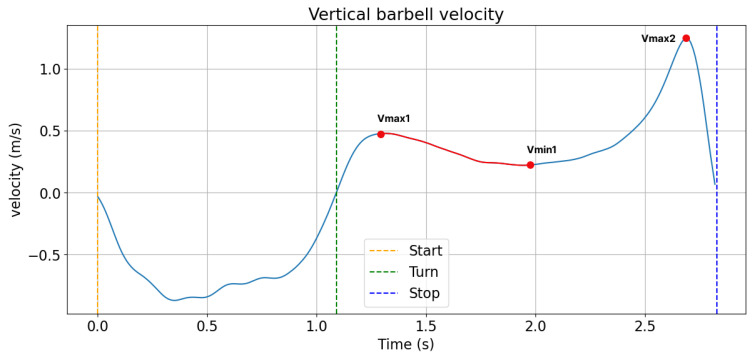
Representative trajectory of one participant’s (male, m = 90 kg, size 1.80 m, additional load 130 kg) vertical barbell velocity during the concentric phase of a maximum or near-maximum 1RM back squat with a pre-sticking, sticking (from Vmax1 to Vmax2), and post-sticking region.

**Figure 2 bioengineering-10-00865-f002:**
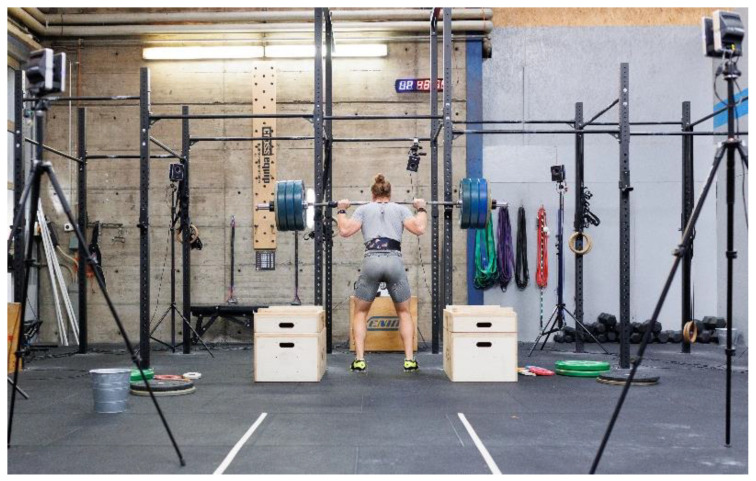
A standard rig with a barbell, weight plates, and safety boxes was used for squat performance testing. Kinematic data of squat performance were acquired using a Vicon Motion System (Vantage 5; Vicon Motion Systems Ltd., Oxford, UK) with ten infrared cameras and two markers on each side of the barbell.

**Figure 3 bioengineering-10-00865-f003:**
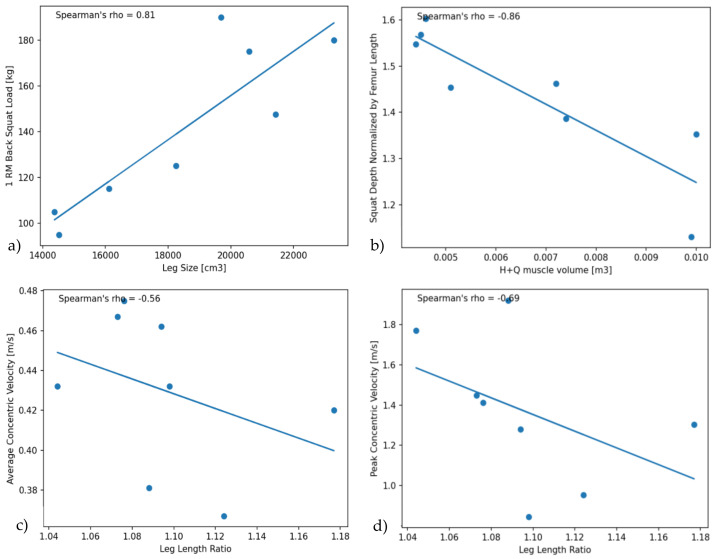
(**a**) Correlation between 1RM back squat load and leg size; (**b**) Correlation between normalized squat depth and the summed hamstrings and quadriceps muscle volume; (**c**) Correlation between peak concentric velocity and leg length ratio; (**d**) Correlation between average concentric velocity and leg length ratio.

**Figure 4 bioengineering-10-00865-f004:**
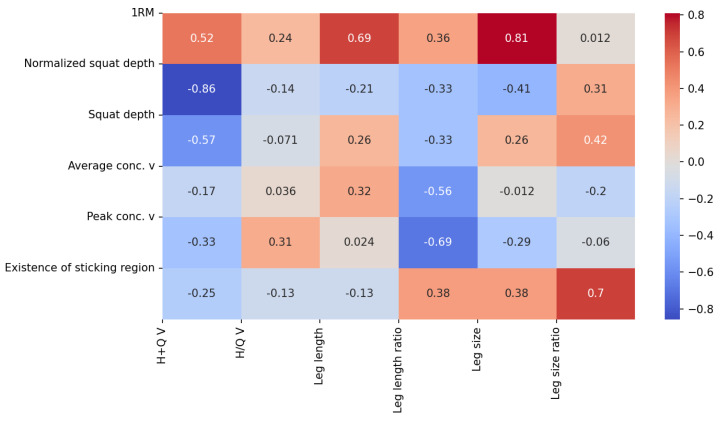
Correlation matrix of the musculoskeletal and squat-specific outcome parameters. Note: v = velocity; V = Volume. Leg length ratio was calculated as the ratio between thigh and shank lengths, while leg size and leg size ratio were calculated from the circumference times the length of the shank and thigh. The values listed represent the respective correlation coefficient (r) values obtained from the Spearman correlation analyses. Following Mukkaka [25], correlation coefficients^®^ were interpreted as low (r = ±0.3–0.5), moderate (r = ±0.5–0.7), or high (r = ±0.7–0.9), respectively.

**Table 1 bioengineering-10-00865-t001:** Musculoskeletal and performance-specific outcome parameters during a set-to-exhaustion at 80% of each individual’s 1RM, including percentage disparity between male and female participants. H: Hamstrings, Q: Quadriceps.

	General (*n* = 8)	Women (*n* = 4)	Men (*n* = 4)	Difference (%)
Age [years]	28.4 (6.46)	29 (7.35)	27.8 (6.5)	−4.5
Height [cm]	172.25 (7.59)	165.63 (0.48)	178.9 (4.1)	7.4
Weight [kg]	80.4 (14.2)	69.1 (10.0)	91.7 (5.3)	24.6
Strength trainings per week	5.4 (2.39)	4.00 (1.63)	6.8 (2.36)	40.7
Strength training experience [years]	7.2 (6.1)	7.4 (8.7)	7.0 (3.2)	5.4
Thigh length [cm]	45 (31)	42 (13)	47 (22)	10.5
Thigh circumference [cm]	61.1 (4.7)	58.3 (4.2)	63.9 (3.5)	8.8
Shank length [cm]	41.0 (3.4)	38.5 (2.4)	43.56 (1.99)	11.6
Shank circumference [cm]	39.4 (3.0)	37.6 (2.5)	41.1 (2.7)	8.5
Hamstring volume 10^3^ [cm^3^]	3.29 (1.35)	2.59 (0.78)	3.99 (1.52)	35.0
Quadriceps volume 10^3^ [cm^3^]	3.35 (1.15)	2.83 (0.60)	3.88 (1.41)	27.1
H + Q volume 10^3^ [cm^3^]	6.64 (2.36)	5.42 (1.37)	7.88 (2.67)	31.2
H/Q volume ratio	0.97 (0.18)	0.91 (0.09)	1.05 (0.23)	13.4
Load (1RM)	142 (37)	110 (13)	173 (18)	24.6
Nr. of repetitions	9.8 (4.4)	8.0 (3.9)	11.5 (4.7)	30.4
Mean squat depth [cm]	64.4 (6.55)	63.9 (6.1)	65.0 (7.86)	1.76
Mean normalized squat depth	1.44 (0.15)	1.5 (0.1)	1.37 (0.19)	9.44
Average concentric velocity [m/s]	0.43 (0.01)	0.42 (0.04)	0.44 (0.04)	3.66
Peak concentric velocity (m/s)	1.37 (0.29)	1.37 (0.34)	1.36 (0.44)	0.29

## Data Availability

Data from this study are unavailable due to ethical restrictions.

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
