# Peer review of "First Insights in the Relationship between Lower Limb Anatomy and Back Squat Performance in Resistance-Trained Males and Females"

_bioengineering, 2023, doi:10.3390/bioengineering10070865_

Round 1

Reviewer 1 Report

The content of the paper is reasonable, there are no comments, suitable for publication.

1. In the manuscript, MRI has been widely used. The novelty and shortcomings of MRI-based muscle volume might be clearly discussed in the manuscript. 2. In the discussion, line 308, it is said that “a clear difference was found in the 1RM back squat load between males and females, which is likely attributed to variances in muscle mass.” Because of the low sample size, the author said that “It appears that the present correlations in the pooled data are exaggerated because the four male participants were both stronger and had larger leg size compared to the four female participants”. There is some confusion in the discussion of the difference between males and females. Due to the small sample size, whether it is appropriate to make such a comparison?

Author Response

Thank you very much for your positive evaluation. 

We were informed that there are additional comments of the reviewer. Thanks for that. Please find our reply below.

  1. In the manuscript, MRI has been widely used. The novelty and
    shortcomings of MRI-based muscle volume might be clearly discussed in
    the manuscript. This has now been included:” Particularly, the acquisition and analysis of MRI data is time-consuming, complicated and costly, thus posing challenges for including more participants. Therefore there are not yet that many studies published in the area of sport sciences using this technology. The quantification of the muscle volume using MRI has also limitations and depends on the position of the human body [27] .”

    2. In the discussion, line 308, it is said that “a clear difference was
    found in the 1RM back squat load between males and females, which is
    likely attributed to variances in muscle mass.” Because of the low
    sample size, the author said that “It appears that the present
    correlations in the pooled data are exaggerated because the four male
    participants were both stronger and had larger leg size compared to the
    four female participants”. There is some confusion in the discussion of
    the difference between males and females. Due to the small sample size,
    whether it is appropriate to make such a comparison?

The discussion has been partially rewritten and reads now the follows:”

Despite the low sample size, the authors consider it noteworthy to look closer at the observed differences between males and females in comparison to existing knowledge in the literature. As expected, the 1RM back squat of the males was larger compared to the  females, which is likely attributed to the larger muscle mass. Indeed, the 1RM back squat load across the pooled data was highly correlated to individual leg size (Fig. 3a), with a moderate positive correlation also found between the 1RM and the summed quadriceps and hamstring muscle volume (Figure 4). Based on this finding, it can be inferred that greater muscle volume is essential to achieve enhanced power potential, and, consequently, a higher 1RM value. Interestingly, recent research reported a correlation between muscle mass and squat depth in female athletes (n=7), whereby performance outcome measures for the parallel or half squat largely depended on the cross sectional area of the vastii muscles but not the hamstrings [26] . Here, it would be interesting to further investigate whether similar correlations are found across female-only, male-only and mixed-gender study groups including participants with a large variation in the muscle mass and the 1 RM performance.” and later in the discussion:”

The main limitation of the present study was the low sample size of n=8, which necessitates care in the interpretation of findings and drawing of preliminary conclusions, especially regarding the resulting differences between genders. Nevertheless, the present sample size is in line with similar studies in the literature that have analyzed muscle architecture in relation to squat performance in vivo [26]. Particularly, the acquisition and analysis of MRI data is time-consuming, complicated and costly, thus posing challenges for including more participants. Therefore there are not yet that many studies published in the area of sport sciences using this technology. The quantification of the muscle volume using MRI has also limitations and depends on the position of the human body [27]. Here, the primary goal of the present study was to provide insights into the relationship between lower limb anatomy and back squat performance across the entire study group, with the analysis of differences between genders being secondary. Further work is needed to confirm the present results and make statistically valid conclusions.”

Reviewer 2 Report

Abstract

·         The hypothesis for this study should be given. The aims should be according to the hypothesis

Introduction

·         “well known” – rephrase without superlatives

·         Figure 1 – what is the source of the presented curve? Is it from other publication (reference and permission should be given). If this is data from the present study – should be in the Results with all the information of the tested individual

·         The hypothesis for this study should be given. The aims should be according to the hypothesis

Results

·         Lines 233-240 and related data in table 1 should be in the Methods

Conclusions

·         should relate to the study hypothesis

Adequate English style

Author Response

Thanks a lot for your valuable comments. 

-It has been added to the abstract:”

It was hypothesized that there is a correlation between subject-specific musculoskeletal and squat-specific parameters.”   -the superlative has been removed.    -The paragraph about the hypothesis has been rewritten and reads now the follows:” Therefore, it was hypothesized that the thigh muscle volume as measure of muscle force potential, assessed by means of MRI, correlates with squat lifting velocity and subject-specific 1RM.  Furthermore, was hypothesized  that the thigh length is inversely correlated to squat lifting velocity, with differences in the correlation expected between males and females similar to the findings of Falch et al. [4] . Additionally, it was hypothesized that there is an association between lower limb anatomy and the presence or absence of a sticking region [22] , whereby the ratio between the length and the circumference of thigh versus shank were considered to play a decisive role.”   - Fig 1 is indeed based on data of this study. The legend has been adapted to:”Figure 1:

Representative trajectory of one participant (male, m=90kg, size 1.80m, additional load 130kg) of vertical barbell velocity during the concentric phase of a maximum or near-maximum 1RM back squat with a pre-sticking, sticking (from Vmax1 to Vmax2) and post-sticking region.“

Furthermore the information about the participants in table 1 has been linked in the result section the follows:”Participants characteristics, including anthropometric measures and muscle volumes from MRI, are given in Table 1.”   -The mentioned lines have been moved to the method section. Also the table 1 has been placed in the method section.    - A part of the conclusion were adapted. Furthermore it has added to the abstract:” It can be inferred that greater muscle volume is essential to achieve enhanced power potential, and, consequently, a higher 1RM value, especially for female athletes that tend to squat deeper than male counterparts.*

Reviewer 3 Report

The paper needs to be revised before publication. Undoubtedly, the major disadvantage is the small study group. I suggests the following improvements (see comments).

1.      L3 - Due to the small number of subjects, it should be clearly added in the title that this is a preliminary or pilot study.

2.      L44 – ‘’ (20F, 4M)’’ - I propose a change of ‘’(20 females, 4 men)’’

3.      L115 –‘’ 2023’’ - add brackets.

4.      L126-128 - The inclusion and exclusion criteria are very unclear. They should be developed.

5.      ‘’ two trainings per week’’ - Add the number of hours spent training.

6.      Add the training experience of the players (in years).

7.      exclusion criteria – ‘’ musculoskeletal complaints’’  - What was the survey like, who conducted it ?

8.      Describe in detail what were the exclusion criteria.

a.      Did the respondents have injuries/problems with the lumbar spine?  Subjects with low back pain have been shown to have different motor strategies.  It could distort the results.

b.      That subjects were tested without pain throughout the body to eliminate its effect on kinematic chains? Please answer in the text with reference to papers 10.26444/aaem/117708 , 10.1519/JSC.0b013e3181d8587b.  If it was otherwise, please give appropriate justification.

9.      Figure 1 – Unify fonts. Enlarge Vmax1, min and Vmax2

10.   Figure 2. - The side photo is unacceptable. You can see the monitor.  The lower limbs are not visible. This should be corrected.

11.   L181 –‘’mm3’’ –‘’ 3’’ - should be in superscript.

12.   L234 –‘’ met all inclusion criteria’’ -  Add the number of all people who signed up for the research.

13.   Table 1. – age –’’ -4.5’’ - Convert to absolute numbers.

Author Response

Thank you very much for your valuable comments.

  1. The title has been changed to “First insights in the Relationship…” _Furthermore it has been stated at line 108: “…the present pilot study… “
  2. This has been changed accordingly.
  3. This issues has been solved.
  4. The paragraph has been rewritten to:” 8 healthy, recreationally active participants were recruited for this study (n=8, 4 male, 4 female). The inclusion criteria were to be healthy, one year of strength training experience with at least one training for one hour per week. Exclusion criteria included musculoskeletal complaints including pain or injury of the lower limbs.”
  5. This information has been added, please see comment 4.
  6. The strength training experience is not presented in the table 1.
  7. The following information has been added to the method section:”The participants had to fill out a questionnaire about the in/exclusion criteria and experience. The questionnaire has been evaluated by the same trained examiner.”
  8. only healthy participants were included. As you mention pain and other issues can alter the movement pattern. Therefore we have only included healthy participants. 
  9. Figure 1 as well as the legend have been adapted accordingly.
  10. Figure 2 right has been removed. It does not add anything for the experimental setup. 
  11. Thanks for noticing. This has been adapted as suggested.
  12. This has been clarified:” 8 participants (4 women and 4 men) signed up for this project. They met all inclusion criteria, signed informed content and enrolled in the study.”
  13. Thanks for this comment. Normally I try to avoid present the same data in the table. Here the difference can be directly calculated based on the other columns. Nevertheless we believe that the difference in % helps to identify parameters with a large difference. This would not be possible if the difference is presented in absolut values. The presentation in % is similar to the concept for normalisation of Hoff, 1996.

Round 2

Reviewer 2 Report

The authors addressed the issues that I suggested for correction

adequate English style